# Floods and Emergency Management: Elaboration of Integral Flood Maps Based on Emergency Calls (112)—Episode of September 2019 (Vega Baja del Segura, Alicante, Spain)

Antonio Oliva * and Jorge Olcina

Department of Regional Geographic Analysis and Physical Geography, University of Alicante, 03690 San Vicente del Raspeig, Spain
* Correspondence: antoniogeografia1@gmail.com

**Abstract:** Emergency mapping makes it possible to manage an emergency situation and even to analyze the catastrophic event, a posteriori, in order to improve action protocols for Civil Protection. The emergency maps are produced from the analysis of calls to the Emergency Coordination Centre (911 or 112). Thus, the concept of integral risk mapping arises, in which risk mapping and aspects that allow for more realistic analysis and mapping through the analysis of emergency calls in the event of a natural event converge. In this case, the analysis is focused on the floods that occurred from 12 to 15 September 2019 in the Vega Baja del Segura district (Alicante, Spain). The results obtained show that this is the flooding episode with the highest number of emergency calls in Valencian region and Vega Baja del Segura district (2010–2022). Likewise, the spatial-temporal analysis of the geolocation of the calls and their reasons, have allowed us to draw up a much more detailed map of flooding or affected areas in 2019 than the official maps. In conclusion, the analysis of emergency calls makes it possible to identify problems and vulnerable areas where proposals can be made to reduce the impact of floods and increase the resilience of a territory. At the same time, it is presented as a novel field of research for the analysis of natural and anthropic risks.

**Keywords:** emergency calls; 911 or 112; integral risk mapping; floods; Vega Baja district



## 1. Introduction

Floods, in their different forms, constitute the natural risk with the greatest socio-economic repercussions on a global scale [1,2]. Since 1980, the increase in exposure due to urbanization and the layout of infrastructures and amenities in floodable areas has been the main cause of the increase in losses related to floods [3]. Other factors have also played a part, such as the inadequate maintenance of flood control infrastructures (damns, channels). In addition, there has been an increase in the hazard risk as a result of the increase in the frequency and intensity of rainfall in certain regions of the world due to the effects of climate change [4–9]. The concentration of assets (human and physical), particularly in urban areas and even more so in coastal areas, which are highly vulnerable to adverse weather conditions, means that the damage is much greater when an extreme weather phenomenon occurs [3].

On a global level, the damage insured for flooding for the period 1991–2000 amounted to 30,000 million dollars; in the period 2001–2010 this doubled, reaching a figure of 68,000 million dollars. Since 2010, an increase has been recorded in insured damage (100,000 million dollars between 2011 and 2021, due, among other factors (economic growth, urbanisation), to the effects of global warming on the development of extreme weather events (frequency and intensity) [10].

Meanwhile, fatalities caused by extreme rain events (floods, tropical cyclones) have also displayed an increasing trend on a global level in recent years, due to the above-

mentioned causes and the effects of climate change (11,497 deaths in the year 2019 and 11,881 in 2021) [3,10,11].

The above circumstances lead to an increasing number of large-scale studies on flood exposure and critical infrastructure and flood assets [12,13].

In Spain, floods are the natural event that causes most catastrophic damage [14,15]. For the period 1987–2006, the compensation (insured damage) due to natural catastrophes amounted to 2,472,594,732 euros, of which, 93.5% corresponded to floods. Only in 2019, the economic losses exceeded 700 million euros [15].

The region of Valencia, on the Spanish Mediterranean coast, is a high risk region in terms of flooding. The river systems of the Mediterranean region are the most hazardous in the whole of Spain due to different factors, such as: the small size and short channel length of the basins, steep slopes, proximity to the coast and a climate with intermittent precipitations and a propensity for torrential rain [2]. Urban agricultural and infraestructure developments are imposing high pressure on the local environment, particulary along the one kilometer wide coastal strip that is dissected by the lowest reaches of the rivers.

The European Directive 60/2007/EC on the assessment and management of flood risks states that floods are a natural hazard that cannot be avoided [13,16]. However, there are a number of measures that can reduce the negative effects of floods, either through structural or non-structural measures. Non-structural measures include those related to prediction, prevention and adaptation, while protective measures are usually based on engineering works. However, at present, such measures have been shown to increase the risk of flooding and its negative effects in the event of failure or breach, while at the same time encouraging greater exposure in flood zones. In this sense, non-structural measures are presented as an alternative to reduce the impacts of floods, through meteorological and hydrological monitoring, land-use planning, flood hazard and risk mapping, land-use regulation, emergency management, flood risk management, among others [16].

The reduction in the natural risk includes a series of measures from prevention to emergency management [16]. One of the most important measures to mitigate the negative effects of the floods and manage an emergency is risk mapping [17]. Mapping is one of the most efficient measures for reducing natural risk [18]. Risk mapping has become an essential tool in natural risk studies and has advanced considerably in recent years due to the application of modern mapping technologies [18].

Risk map elaboration is currently carried out using Geographic Information Systems (GIS). These are tools that enable the acquisition, storage and edition of spatial information. In other words, they enable us to create a georeferenced database or integrate data into an existing one [19]. GISs allow us to carry out different risk analysis operations and procedures and are highly useful for designing and implementing mitigation measures [1].

Emergency mapping forms part of preventive culture, understood as a common behaviour of society, learnt to assertively respond to hazardous situations that may arise [20]. It facilitates the management of an emergency situation and even the analysis of the catastrophic event a posteriori, in order to improve Civil Protection action protocols.

One example of emergency maps is those based on the analysis of calls to the Emergency Coordination Centre (911 in USA and 112 in the European Union) after an event has occurred, to assist with the design of evacuation routes for vulnerable populations of for emergency responders to gain access to affected areas [1,21,22].

After an event and depending on the scale of the phenomenon, an analysis process can be initiated due to an increase in knowledge which will help to enhance resilience through a greater adaptation to the risk [23]. This is how the comprehensive risk map concept has emerged, in which risk mapping is blended with aspects that contribute to improving risk prevention and resilience based on the data obtained in the maps that reflect the real situation of the areas affected by an event, including those elaborated through emergency call analysis in a territory affected by an extreme natural event. Comprehensive risk maps add a greater level of reality in territorial planning and emergency management processes (Figure 1).

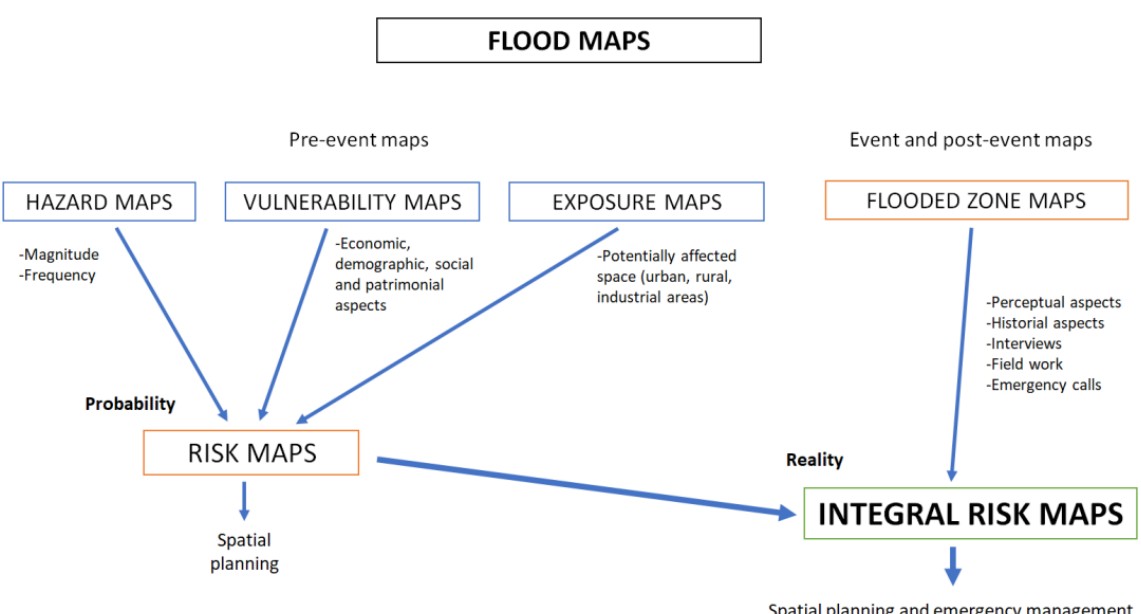

**Figure 1.** Flood maps types. Own elaboration.

The number for emergency calls in Spain is 112 or 1-1-2. The call is received at the Emergency Coordination Centre of the Valencian Community. Calls to provide highly representative information about where and when an incident is occurring and the type of incident. In the majority of flood cases they are related to functional disruptions of everyday life and economic activity, generated initially by the rain in situ [22] and pluvial flooding; and, subsequently, depending on the response of the river basins, by fluvial flooding.

Rossi et al. (2013) indicate three main reasons why the emergency data have sufficient spatial resolution for study or analytical purposes. First, the emergency calls are made in situ. In other words, where the incident is occurring. Unless affected by an extreme phenomenon, a call can be made in a flooded basement, for example. Second, precise locational information should be given by the caller to the emergency co-ordination center and then to the rescue authorities. In the case where the location is unknown, the emergency coordination centre traces and locates the place of the incident. Third, and finally, the authorities act based on this emergency report [23].

One of the problems of emergency calls is that they are classified according to the type of incident. However, as this is conducted in real time, one type of category may be confused with another. For example, a flood may be occurring due to a reactivation of a river channel, but it is classified as a pluvial flood [23].

Few studies analyze emergency calls. The records of these calls constitute an important source of information for understanding the processes of the evolution of storms and the natural dynamics of the territory. However, they are undervalued but have enormous applicability in a whole range of situations and risk analysis (natural and human).

Emergency calls allow the identification of spaces vulnerable or prone to a natural event in relation to the physical conditions of the terrain. They also make it possible to identify watercourses that are not mapped on official maps.

They also enable the impact of the event and its consequences to be determined together with the functionality of the road infrastructures and access routes for emergency teams. In short, they improve our understanding of the functioning of the natural dynamics of a territory in a situation of a natural event. Their analysis enables us to take preventive and management measures during an emergency and can be applied to the emergency management of intervention teams.

In the field of medicine and hospital emergencies, 112 calls fulfil an essential function [21]. Furthermore, their use for studying natural risks, including floods, is a field of analysis yet to be explored [23].

Many studies have used emergency calls as additional information in analysing an event [21–29] (Table 1). Their results are excellent in terms of risk analysis because they efficiently complete the results of risk mapping.

**Table 1.** International and national studies that have used emergency calls.

| Scale | Publication | Description |
|---|---|---|
| **International** | Integrating social media in Emergency Dispatch via Distributed Sensemaking [21] | Use social network resources, such as tweets, which allows to know what is happening in real time through the description of the perpetrator, photographs or videos; at the same time it allows to interact with other users from other nearby areas who can pick up the real situation in an exact point, at a specific time, which can serve as a preventive measure or warning in the development of the emergency |
| | Real-Time hazard approximation of long-lasting convective storms using emergency data [23] | Analyse the damage produced by hail using the recorded emergency calls, the information of the insurance companies and weather radar data. |
| | Sydney hailstorm based on ground observations, weather radar, insurance data and emergency calls [24] | Use emergency calls reports and insurance data to quantify the scope of the losses due to intense rainfall |
| | Quantifying the risk of heavy rain: Its contribution to damage in urban areas [25] | Use online emergency call reports as an additional source of information, together with weather radars and the evolution of a convective storm |
| | Using 311-Call data to Measure Flood Risk and Impacts: the case of Harris Country TX [26] | Spatially and descriptively examine 19,680 emergency calls made during hurricane Harvey (2017) in Houston, Texas. Their results indicate the usefulness of these data to identify the flood risk and to mitigate the growing costs of floods more proactively |
| | Using Geospatial Technology to process 911 Calls after Hurricanes Katrina and Rita [27] | Analyse how the USGS used emergency calls after the Katrina hurricane (2005) so as to be able to send emergency teams to rescue people affected by the floods |
| **National (Spain)** | Lluvias in situ en la Comunidad Valenciana. Relación entre indicadores pluviométricos, llamadas al centro de coordinación de emergencias (112) y relación de daños durante el episodio de 26–30 de noviembre de 2016 [22] | Use emergency calls, together with rainfall and hydrologic information to identify pluvial flood problems in situ, particularly on the roads in the Region of Valencia, analysing the episode of 26–30 November 2016 |
| | Can the Quality of the Potential Flood Risk Maps be Evaluated? A case Study of the Social Risks of Floods in Central Spain [28] | Seeks to gauge and validate flood risk maps at different scales in the autonomous region of Castilla-La Mancha (Spain), establishing a relationship between the number of calls to the emergency services and the categorised level of flood risk derived from the PRICAM project. The results of the research are positive and show the usefulness of applying these data to the study of risk, in this case, of floods |
| | Advancing in flood risk communication: a proposal for early-warning messaging [29] | Analyses the total number of emergency calls to create a methodology and proposed library of messages for use in SMS |

Note: Source: Own elaboration.

The use of emergency calls is gaining prominence in the analysis of flood episodes and post-catastrophe scenarios with the elaboration of territorial action plans, such as the Plan Vega Renhace, with the primary objective of creating a library of messages by analysing the calls to the emergency services. In order to create this message library, a total of 14,194 calls have been analysed, made between 10 and 20 September 2019, of which 4078 corresponded to the afore-mentioned area [29].

This research analyses the flood episode in the district of Vega Baja del Segura (Alicante, Spain), using emergency calls made to the Emergency Coordination Centre of the Region of Valencia between 12 and 15 September 2019.

During this period, the formation of a *gota fria* (cold drop) atmospheric situation in the Algiers sea favoured the formation of convective storms headed towards the district of Vega Baja del Segura, generating a first flash flood on the 12 September at 10:00, releasing 250–300 L/m$^2$ and causing major problems for homes and roads while reactivating many river courses. On 13 September, the largest rivers overflowed causing the destruction of several of the hillocks of the River Segura, completely flooding the Vega Baja district. The land was covered with water for approximately a month in some municipalities, as there is hardly any gradient, among other factors, such as an inefficient and obstructed river mouth.

The objectives of this study are:

(a)   To analyse the emergency calls made between 12 and 15 September 2019 related to the flooding event.
(b)   To elaborate a flood map of the episode of 2019 in the district of Vega Baja del Segura based on the emergency calls.
(c)   To identify floodable areas that are not contemplated in the official maps and which should be incorporated as high risk areas due to the real effects suffered after an important flooding event.
(d)   To respond to the following question: Are the 112 calls more related to the flooded area or the potentially floodable areas of the flood maps?

## 2. Materials and Methods

### 2.1. Study Area and Data

The Region of Valencia is the Spanish region which suffers from the greatest economic and social losses due to flooding [30]. The assets insured by the flooding episode of September 2019 amounted to €181,791,150, of which, €171,186,533 corresponded to the district of Vega Baja [31] which represents 94.17% of the total economic loss. This illustrates the size of the flood episode and identifies the enormous problems of exposure to floods in the district as it occupies the floodplains of the river Segura and other principal rivers, such as the Abanilla, Salada, Algüeda, El Derramador, Siete Higueras, Pilar, among others.

The study area comprises municipalities that belong to two subregional units (districts): the Vega Baja del Segura and the Bajo Vinalopó. Both are in the south of the Region of Valencia. The first district covers the whole of the territory called La Vega Baja del Segura, located in the southernmost sector of the Region of Valencia. This district is made up of a total of 27 municipalities, whose total population amounts to 350,000 inhabitants, rising to one million inhabitants in the summer months, which implies a greater exposure and vulnerability to the natural threat of floods. Of the second district, the municipalities of Crevillente and Elche have been chosen, which were also affected by this flood episode. The waters of the different rivers came down from the Crevillente sierra to the Vega Baja del Segura, generating floods in hamlets and other land uses in the floodplains of the river Segura. This justifies the analysis of these two municipalities. The area of study covers, in total, 29 municipalities that were seriously affected by the flood episode of September 2019 (27 in Vega Baja del Segura and 2 in Bajo Vinalopó) (Figure 2).

This study uses data from emergency calls made to 1·1·2 or 112 registered in the Emergency Coordination Centre of the Region of Valencia. The call data have been provided by the Valencian Agency for Security and Emergency Response (AVSRE) of the Regional Government of Valencia.

The general search criteria or data selection for their analysis correspond to an extraordinary rainfall emergency, for the period between 10 (00:00:00 h) to 15 September 2019 (23:59:59). These dates were when the two flash floods occurred (12 and 13) and the more serious flooding situation took place (13–15 September 2019). The maps to be represented correspond to day 12 and 13. Subsequently, a final map is represented with the totality of calls from day 12 to 15.

The emergency calls are classified into a principal field which is subdivided into different types, differentiating between urgencies and emergencies, to transmit to the intervention team (Table 2).

**Table 2.** Classification of the types of calls registered in the Emergency Coordination Centre (112).

| | | | |
|---|---|---|---|
| **Types of calls** | **Accident** | Vehicle | *Trapped* |
| | | | *Unknown injured* |
| | | | *Injured* |
| | | | *Fire Injured* |
| | | | *Fire No Injuries/Desc.* |
| | | | *MMPP* |
| | | | *No injuries* |
| | **Natural Phenomenon** | Rain | *Water Drainage/Filtration* |
| | | | *Water Immovable* |
| | | | *Overflowing Watercourse* |
| | | | *Other* |
| | | | *Flooded Livestock Way* |
| | | | *Vehicle blocked* |
| | **Incident** | Trafficc | *Railway* |
| | | | *Road Network* |
| | | Material Damage | *Official Building* |
| | | | *Property Injured* |
| | | | *Building Not in Danger* |
| | | | *Property/Building* |
| | | | *Public Road* |
| | | | *Public Road Injured* |
| | **Rescue** | Search | *Rural/Mountainous* |
| | | Landslide | *Building/Construction* |
| | | | *Land* |
| | | Rescue | *Inland Water* |
| | | | *Cave/Sima* |
| | | | *Rainfall* |
| | | | *Rural/Mountainous* |
| | **Service** | Social | *Humanitarian* |
| | | | *Minor* |
| | | | *Other* |
| | **Bassic Service** | Water | *Wide Area Cut* |
| | | Electricity | *Cutting Wide Area* |
| | | | *Cut Supply* |
| | | | *Installation Damage* |
| | | Gas | *Cutting Wide Area* |
| | | Telephone | *Cutting Wide Area* |

Note: Source: AVSRE.

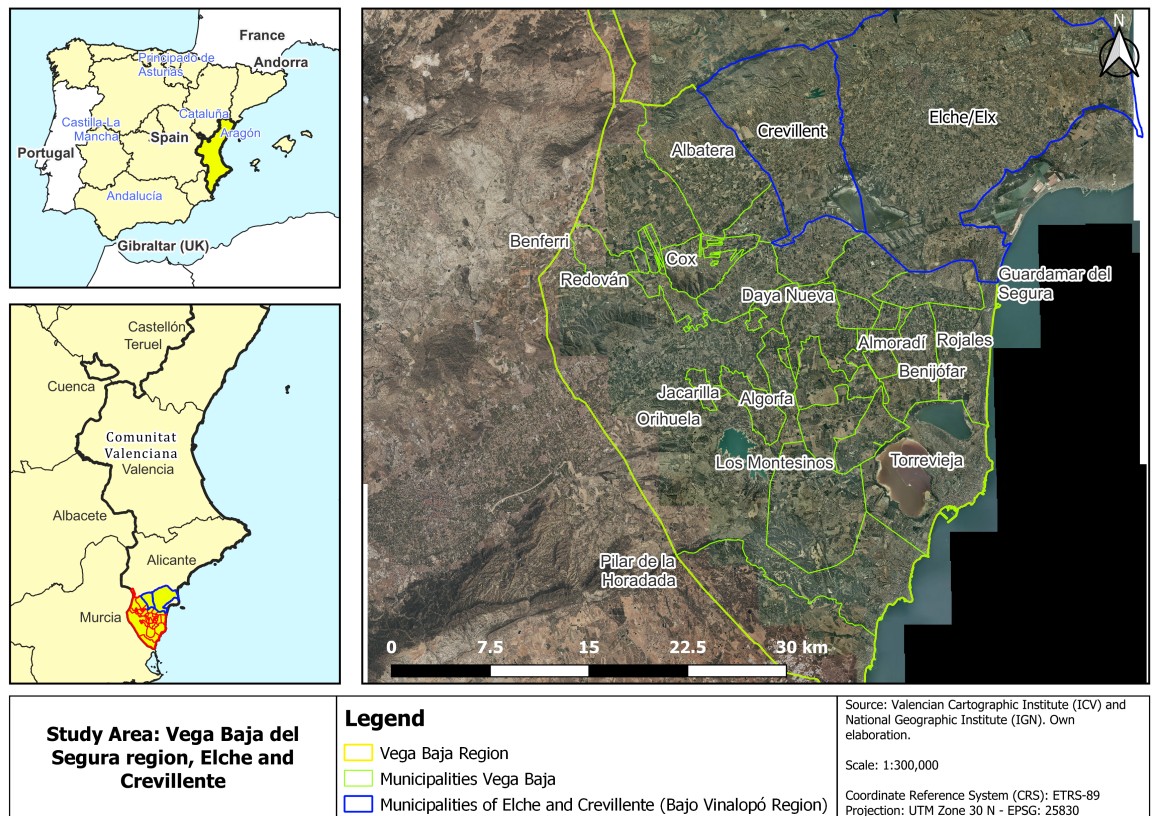

**Figure 2.** Study Area: Vega Baja del Segura, Elche and Crevillente. Source: Valencian Cartographic Institute (ICV) and National Geographic Institute. Own elaboration.

The data used have had to be filtered in order to reflect the reality associated with the flood. Taking into account that the episode analysed began on the 12 September, the calls on the previous days (10 and 11) have been discarded. Those emergency calls that were not directly or indirectly associated with the flood episode have also been excluded.

In short, a total of 3648 calls to 112 have been analysed between 12 and 15 September, both inclusive, in the area of study (Figure 3).

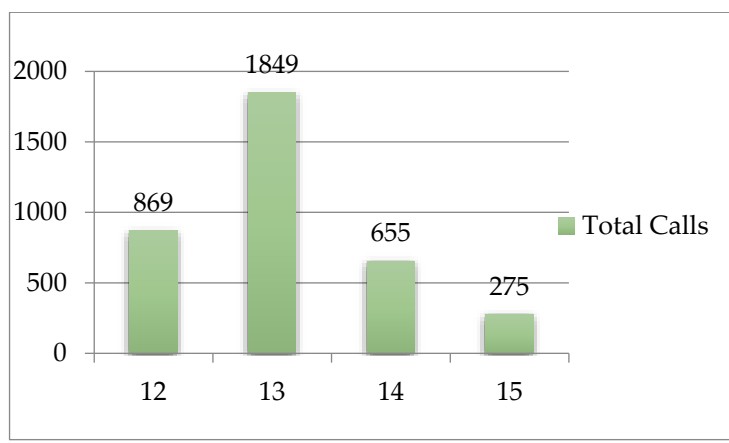

**Figure 3.** Total calls to the Emergency Coordination Centre (112) 12–15 September 2019. Source: AVSRE. Own elaboration.

Table 3 shows the number of calls made to 112 registered by municipality in the area of study, where Orihuela stands out, given the enormous size of the municipality (443,

2 km$^2$). This implies a greater number of susceptible and vulnerable areas to a flood episode (Table 3).

**Table 3.** Total calls by county and municipalities. 12 to 15 September 2019.

| Province/County/Municipality | Total Calls (112) |
|---|---|
| **El Bajo Segura** | **3424** |
| Albatera | 18 |
| Algorfa | 26 |
| Almoradí | 209 |
| Benejúzar | 16 |
| Benferri | 101 |
| Benijófar | 14 |
| Bigastro | 106 |
| Callosa de Segura | 47 |
| Catral | 141 |
| Cox | 44 |
| Daya Nueva | 25 |
| Daya Vieja | 9 |
| Dolores | 198 |
| Formentera del Segura | 4 |
| Granja de Rocamora | 41 |
| Guardamar del Segura | 26 |
| Jacarilla | 31 |
| Los Montesinos | 7 |
| Orihuela | 1839 |
| Pilar de la Horadada | 127 |
| Rafal | 48 |
| Redován | 108 |
| Rojales | 11 |
| San Fulgencio | 20 |
| San Isidro | 47 |
| San Miguel de Salinas | 19 |
| Torrevieja | 142 |
| **El Bajo Vinalopó** | **311** |
| Crevillent | 48 |
| Elche/Elx | 263 |
| **Total count** | **3648** |

Note: Source: AVSRE. Own elaboration.

Once the calls had been filtered and classified, each hour of each day was analysed, with the aim of identifying flood zones in urban areas, on the flood plain and the reactivation, overflow and flooding of the main river courses. The results are reflected in the mapping carried out.

## 2.2. Geolocation of the Calls to 112

The analysis of the emergency calls revealed that were projected on geographic coordinates (X, Y). Therefore it was decided to transform them to the UTM Zone 30 N, EPSG: 25,830 projection, as stipulated by the European regulations for mapping.

In order to enter the data into a GIS, a database was previously created in Excel (.xml) format so that the file could be subsequently transformed into CSV code (comma-separated values). This transformation is necessary so that the columns of the database created can be read correctly by the GIS.

The GIS used is the QGIS (3.26.2 Buenos Aires) to which CSV layers were added. After this layer was added it had a shapefile layer format (.shp). This enabled us to locate the calls in the area of study and understand the flood process of the event occurring in September 2019.

The analysis includes a detailed study of the calls for each hour and the type of incident from 12 to 15. This makes it possible to reconstruct the evolutionary process of torrential rainfall, the reactivation of ravines, the first areas that begin to have flooding problems, the overflowing of watercourses, among other problems. Generally, the number of calls is higher during the first hours of sunshine (06:00 a.m.) with the start of the working and school day until 22:00 p.m. (22:00 p.m.). The analysis of the number of hours per day makes it possible to reconstruct each day's events and, as a whole (12–15), to understand the evolution of the disaster in a given territory.

However, due to restrictions of space for this article, the results are shown on a district level and total calls per day. However, all these data have been analysed in order to understand the dynamic fluvial processes existing in Vega Baja del Segura.

## 2.3. Flood Mapping in Relation to the 112 Calls

The geolocation of the calls and their analysis has enabled us to identify the causes and consequences of the floods. In other words, the causes of the floods have been identified in detail, highlighting streets that act as channels, current courses, historical channels and principal river courses that cause the floods, and, therefore, the calls to 112.

One of the results that is shown and analysed in its corresponding section is that many calls are made from areas which the official maps (Sistema Nacional de Cartografías de Zonas Inundables (SNCZI) (https://sig.mapama.gob.es/snczi/) (accessed: 12 December 2022) and Plan de Acción Territorial de Riesgo de Inundación de la Comunidad Valenciana (PATRICOVA) (https://visor.gva.es/visor/) (accessed: 12 December 2022) do not identify in terms of fluvial hazard. Therefore, determining the flooded areas through the emergency calls allows us to complete these maps. One of the closest layers with respect to the 2019 flood can be explained if the geomorphological hazard of the PATRICOVA is included. This layer is not included in the SNCZI.

Another problem that the analysis of the emergency calls has identified is that there are geolocalised points in areas where no type of fluvial channel is currently observed. In order to resolve this problem and to understand the situation prior to the current context, the location of the calls to 112 with respect to the episode of 2019 have been analysed together with aerial photographs of the American Flight Series B (1956). This analysis has verified the existence of former channels which today have been transformed by current uses which explains their reactivation and the generation of floods.

After analysing the situation in 1956, the fluvial channels that were reactivated in the flood of September 2019 were identified and mapped. Subsequently, and taking into account the characteristics of the terrain and the area of study and behaviour of the water on the terrain, a polygon map has been elaborated of the flooded areas with the activity of the channels that were reactivated with the torrential rain.

The final result of using this methodology has enabled the elaboration of a "real" map of the flooded areas in 2019. This map can be considered as a fluvial hazard and geomorphological map which could become a "real" flood risk map of the episode of 2019. The analysis of this episode based on the calls to 112 has enabled us to identify flooded areas,

learn about the dynamics and functioning of the channels of the area of study, elaborate a hazard map and a "real" map of the flooded areas in 2019 and, therefore, will allow us to manage future emergencies on a district and local level, safeguarding human lives. Finally, it also allows us to identify problems for which comprehensive solutions can be designed (structural and non-structural).

## 3. Results

In order to determine the spatial and time distribution of the emergency calls in the area of study, we first need to examine the evolution and development of the flood event of September 2019.

### 3.1. Principal Landmarks of the Flood of September 2019

According to the information provided by the Automatic Hydrological Information System of the Segura River Basin, more than 500 mm were recorded in just 36 h in the munic­ipality of Orihuela. The maximum torrential rain discharges occurred on the 12 September from 11:00 a.m. to 13:00 p.m. (Situation 1); and on the 13 September, starting with greater force from 7:30 a.m. to 11:00 a.m. (Situation 2). Emergency calls to 112 started to be registered a little before the two strongest flash floods began. In general terms, we can dif­ferentiate that in the first flash flood (Situation 1) the majority of calls referred to problems with the rainwater and fluvial floods. The second flash flood (Situation 2) led to an increase in the number of calls related to overflowing and flooding rivers (Figure 4).

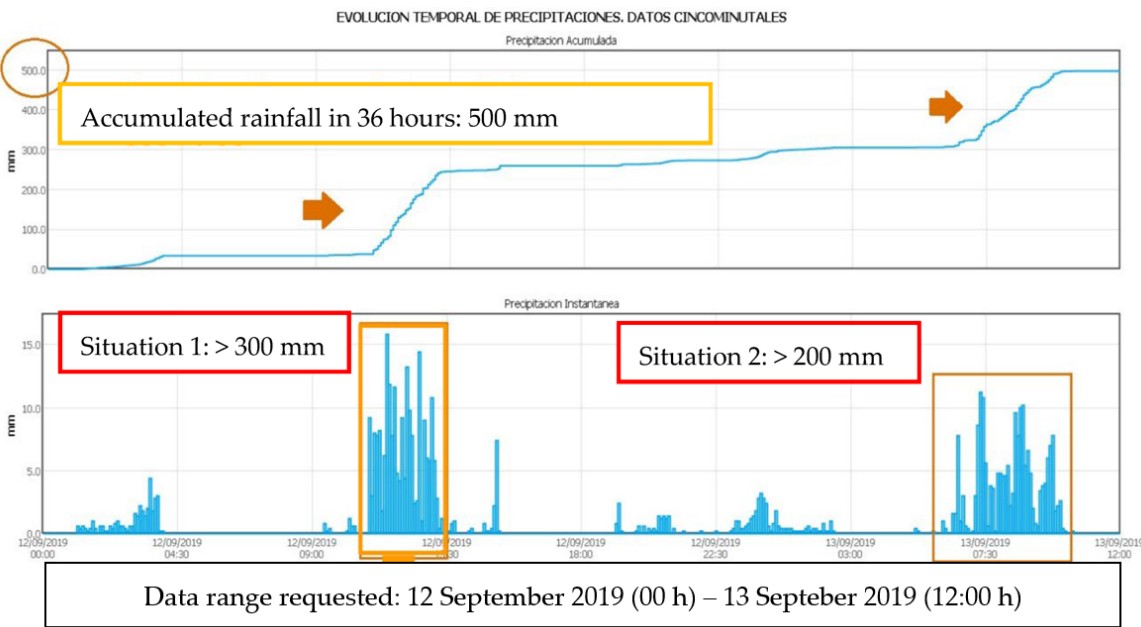

**Figure 4.** Temporal evolution of precipitation. Five-monthly data (**top**) and instantaneous precipita­tion (**bottom**). Source: SAIH. CHS.

These torrential rains discharged in the indicated timeframe led to the reactivation of many principal fluvial channels, generating floods in many parts of the area of study with an exponential increase in the number of 112 calls. The following figure summarises the principal events of the flood episode of the 12–15 September 2019 in the area of study (Figure 5).

Figure 5 shows the chronology of events and the most significant territorial effects in the September 2019 flooding episode. In relation to the emergency calls (112), it should be noted that it is impossible to know the individual effects of each of the calls. Sometimes several calls come in at the same time and turn out to be the same problem. However, the AVSRE database, in addition to the classification of the calls, includes a brief description of where the call occurs and what is happening at that moment. In this line, in summary,

the effects of the flooding reported in the 112 calls have to do with: rescue and evacuation of people, flooded houses, flooded basements and garages, roads cut off by the flow of watercourses, urban streets that act as rivers due to steep slopes, landslides and slides of rocks from the mountains, house fronts, among many others. Many of the affected areas were flooded with water depths in excess of 0.5 m and over 0.80 m.

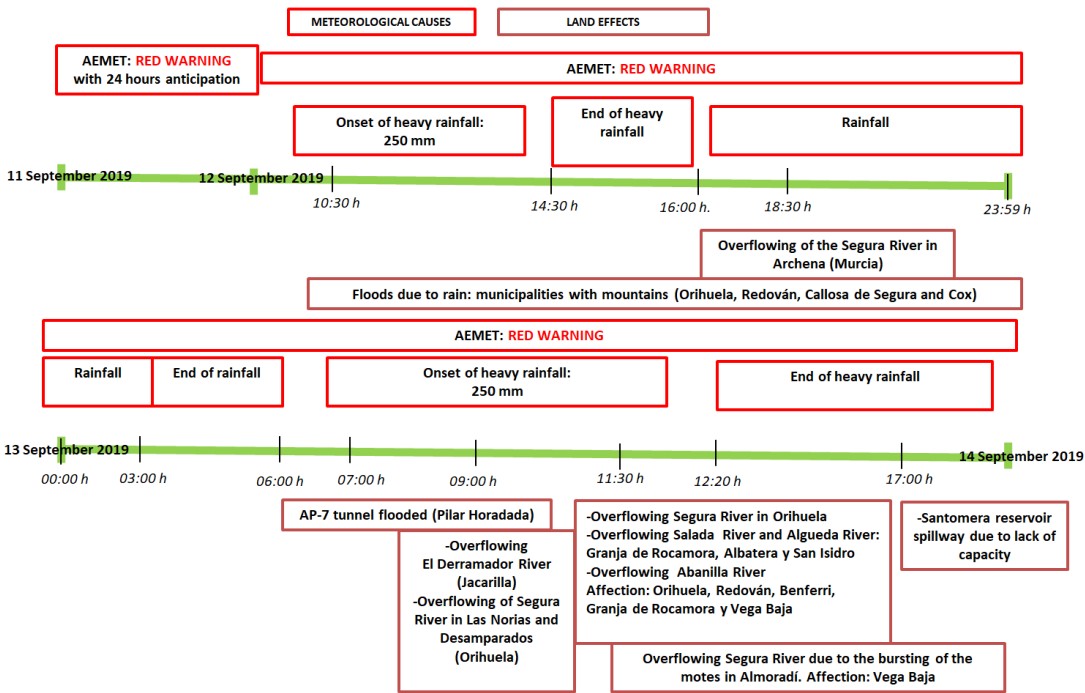

**Figure 5.** Chronology of main events 12 and 13 September 2019. Own elaboration based on information from 112 calls.

The reaction or response to calls to 112 made the emergency teams (local, provincial, regional and state) available. The emergency devices that intervened were the land and marine Military Emergency Unit (UME), the State security forces (local and national police), firefighters (local, provincial and regional), members and volunteers of Civil Protection from the affected municipalities and other parts of the province of Alicante and Spain, and volunteers from the area or other localities who offered help and shelter for the people affected and evacuated. Centres or shelter points were set up where the affected population could go and where they had prepared a large enclosure with beds, blankets, hot food, water, clean clothes, among other services. Once the people had been evacuated, the emergency services carried out animal rescue tasks, which could have been saved if they had been considered beforehand, in the management of a flood zone, with the aim of safeguarding them.

Another of the effects of the flooding episode is related to mental health. There is no doubt that the September 2019 episode left significant after-effects on the affected population, generating fear, insecurity and trauma. Some people's feet felt wet days after the flooding had passed or they were afraid of a warning of rain. This highlighted the importance of having mental health specialists (psychologists) available to support those affected during the event and beyond.

There was room for improvement in the management of the flooding. Faced with a red warning for rainfall decreed 24 h in advance, the local councils that had a Civil Protection plan (emergencies and floods) activated the so-called CECOPAL (Local Administration Operational Coordination Centre). This meant that the local councils were concerned with

the resources they had available and with attending to their problems, without being able to send help to other municipalities which, due to their size, population and scarcity of resources, have few personnel from the State security forces or a municipal Civil Protection corps. Given that the municipal resources were exceeded in all the municipalities, a higher plan had to be activated, the regional CECOPAL, to take charge of the management of the emergency in a planned and coordinated manner, trying to distribute resources and means, while at the same time attempting to respond to the 112 calls that were received.

In addition to the problem of not having Civil Protection Plans for emergencies in most of the municipalities in the study area, there is also a lack of a Plan superior to the municipal one, on a regional scale, which understands the functioning of the region and the possibility of being affected throughout the territory by a single agent (Segura river, Abanilla river, El Derramador river, Salada river, among others). Therefore, there is a need to draw up a regional flood emergency plan for the Vega Baja region.

After the flooding, the Valencian regional government promoted a Strategic Plan for adaptation to atmospheric extremes and climate change, with the aim of making the Vega Baja del Segura region more resilient. In this line, the "Plan Vega Renhace" arises from public participation and collaboration between the different administrations (local, regional and national), as well as incorporating businessmen and associations [32].

The pillars of the Plan Vega Renhace are classified into 4 main lines of action: hydraulic infrastructures, climatic emergency, economic development and society [32]. In the second axis of "climatic emergency", it was possible to incorporate the drafting of Civil Protection Emergency and Flood Plans for the local area. The next step is to elaborate a regional plan that includes all the issues mentioned above.

### 3.2. Analysis of the Emergency Calls in the Region of Valencia, Province of Alicante and Area of Study (2010–2022)

The Emergency Coordination Centre (112) registered a total of 25,523 calls directly or indirectly related to the floods (pluvial and fluvial) in the Region of Valencia in the time series 2010–2022 (Figure 6). The trend shows an exponential increase in the number of calls related to floods in the Region of Valencia from the year 2010 to the year 2022. Particularly noteworthy are the years 2012 (2748), 2016 (2389) and 2019 (7433), corresponding to flood episodes in the Segura river basin and, therefore, in the chosen area of study. This Is due to a greater exposure and, therefore, occupation of the population and economic activities in floodable areas in the autonomous region (Figure 6).

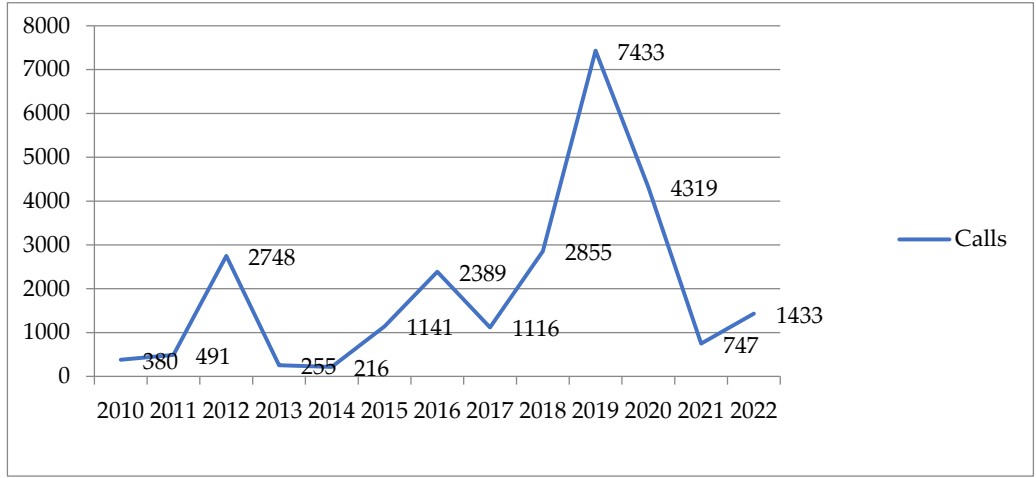

**Figure 6.** Emergency calls registered at the Emergency Coordination Centre in the Comunidad Valenciana in relation to floods (2010–2022). Source: AVSRE (2022).

Of the 25,523 calls registered in the CCE in the whole of the Region of Valencia for the period 2010–2022, 9474 corresponded to the province of Alicante, where the most

noteworthy years were 2012, 2017 and 2019. With respect to the number of calls registered in the province of Alicante, the Vega Baja district recorded a total of 6242 emergency calls in the time series 2010–2022, where 90% corresponded to the torrential rain episode of 2019, followed by 2016 and 2012 (Figure 7).

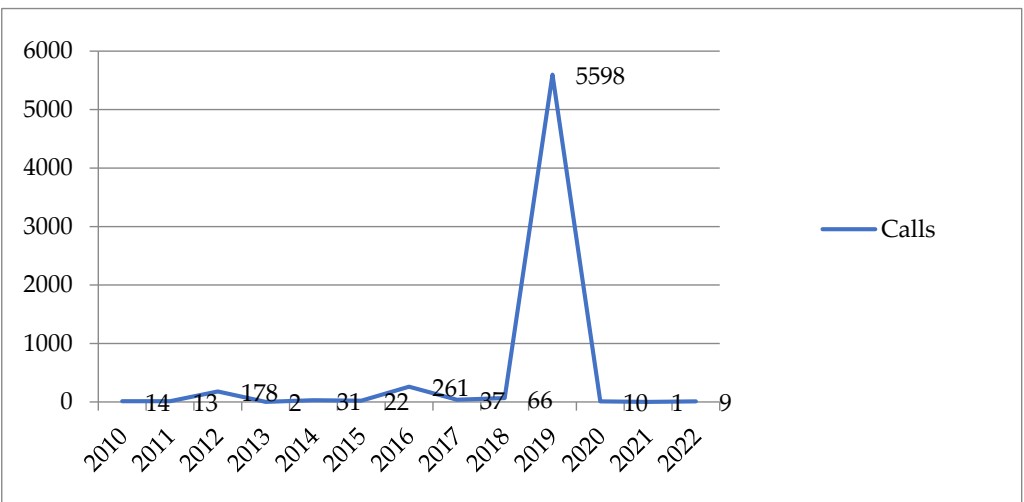

**Figure 7.** Emergency calls registered at the Emergency Coordination Centre in the Vega Baja del Segura district in relation to floods (2010–2022). Source: AVSRE (2022).

The municipalities of the district of Bajo Vinalopó included in the area of study (Elche and Crevillente), made a total of 523 calls, 50% of which were registered in the episode of September 2019 (Figure 8).

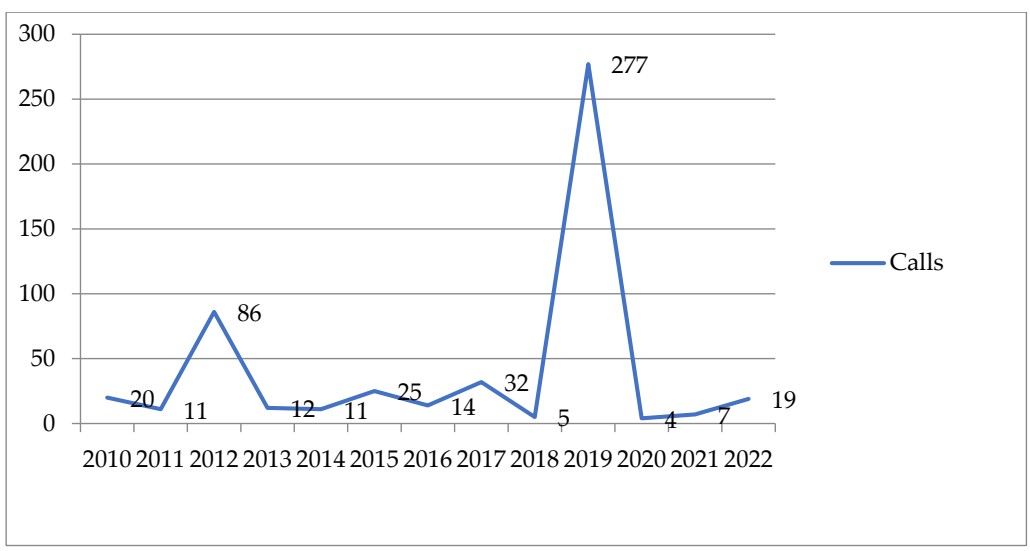

**Figure 8.** Emergency calls registered at the Emergency Coordination Centre in Elche and Crevillente (Bajo Vinalopó district) in relation to floods (2010–2022). Source: AVSRE (2022).

In both regions, emergency calls are unusually high due to a greater exposure and occupation of flood zones which, in situations of extreme meteorological phenomena, are seriously affected.

*3.3. Space-Time Distribution of the Emergency Calls and Flooded Areas in the District of Vega Baja del Segura and Elche and Crevillente (12 to 15 September 2019)*

The emergency calls allow us to analyse in depth the dynamics of the territory and how they functioned in the district of Vega Baja del Segura in the flood episode of September

2019, enabling us to understand their functioning in the case of a similar situation in the future.

The analysis conducted has allowed us to determine the location of the call, the reason for it, the space-time distribution of the problems and the evolution of the floods. It has enabled us to study in detail the space-time distribution by hour. However, the maps shown below summarise the total calls made on 12–15 September 2019.

The location of the calls enables us to identify the fluvial courses (rivers, watercourses, ravines, ditches, etc.) which began to function and caused flooding problems in urban centres and the floodplain. In some cases, due to the intense transformation of the territory, the defined channels had disappeared and it seemed as though they had never existed. However, this is a false perception as in a situation of heavy and intense rains, these channels recover their function, affecting human uses that previously did not exist. In order to identify the existence of these channels, the American Flight Series B of the year 1956 has been used in which they can be found.

After identifying the channels, a polygon map was elaborated of the flooded areas. Two large groups have been identified: the channels with their own and sufficient entity and floods by municipality, which includes fluvial channels that were reactivated and affected the urban centres.

The first group is made up of the principal channels of the district which flood large areas of land. In this first group we can highlight: the floods caused by ruptures and overflows of the river Segura, the floods caused by the reactivation of the Abanilla watercourse, the overflowing of the Azarbe Mayor de Hurchillo, the reactivation of gullies and the flooding of the watercourse between Jacarilla-San Miguel de Salinas-Los Montesinos, the reactivation of the watercourse of Algueda and the flooding of the Salada watercourse.

The second group includes floods occurring in different municipalities due to different fluvial channels. Furthermore, areas flooded due to other fluvial channels are present that are not included in the map legend but where emergency calls are located. We should also remember how the municipalities of Elche and Crevillente belonging to the district of Bajo Vinalopó were affected, although a large part of their fluvial channels descend towards the district of Vega Baja del Segura, affecting this area.

Figure 9 shows the emergency calls registered on 12 of September 2019, together with the total flood map. As previously mentioned in Section 3.1, the calls registered on the 12 September particularly refer to pluvial floods due to the discharge of high amounts of rain and its torrential nature (+250–300 L/m$^2$). This explains why the majority of calls were made in the foothills of the Orihuela and Callosa de Segura Sierras. As we can observe in Figure 8, the town of Orihuela registered many calls, as did the whole of the southern and northern face of the Orihuela Sierra. Meanwhile, the emergency calls made in the villages of Redován, Callosa de Segura and Cox corresponded to the edge of the Callosa de Segura Sierra. The number of calls made from Rafal, Almoradí or Torrevieja match the insufficient drainage system existing in these areas and the rainfall generate considerable flooding problems.

Elche is the municipality of the district of Vinalopó that registered the highest number of calls on 12 September associated with the reactivation of some ravines descending from NW to SE.

Figure 10 shows the total emergency calls registered on 13 September 2019, together with the total flood map. On this day, another 250–300 mm were discharged, practically constituting a historical record of more than 500 mm. This was the worst day of the flood event as the majority of the fluvial channels of the district were activated. This is when the final map of flooded areas began to be drawn. We can observe that the majority of calls were concentrated in the town of Orihuela (overflowing of the river Segura), between the Orihuela and Callosa Sierras, due to the reactivation of the Abanilla watercourse and in Almoradí-Dolores-Daya Nueva due to the overflowing of the river Segura caused by the rupture of the channels of the river Segura.

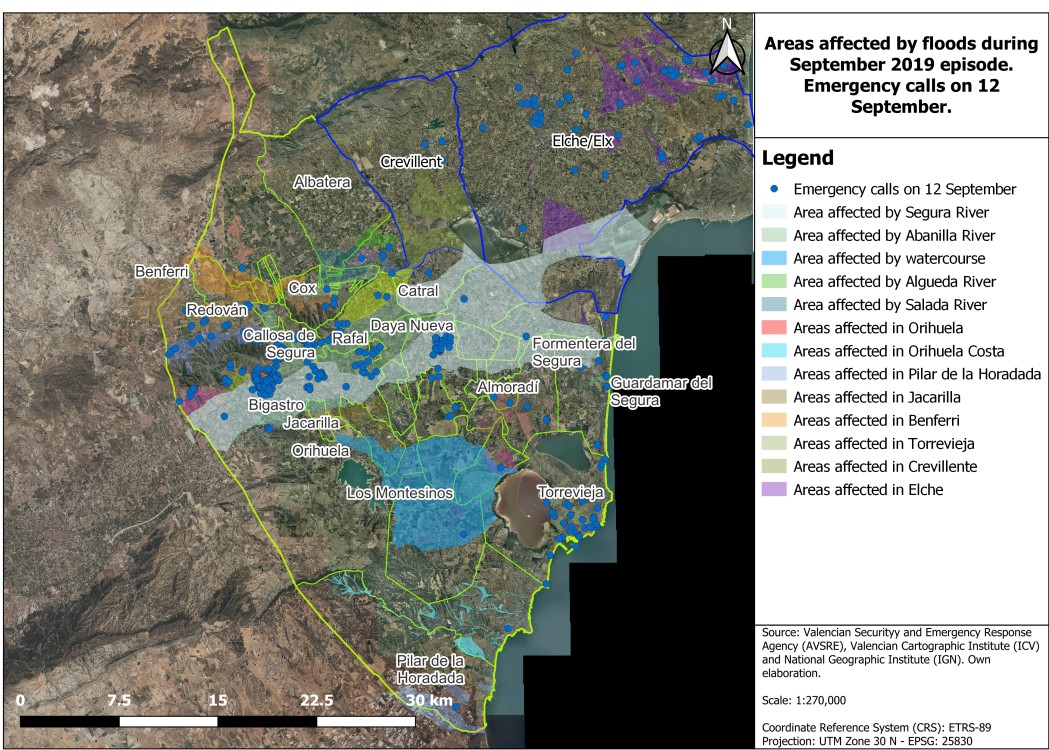

**Figure 9.** Areas affected by floods during September 2019 episode. Emergency calls on 12 September. Source: Own elaboration based on calls to the Emergency Coordination Centre (CCE). AVSRE.

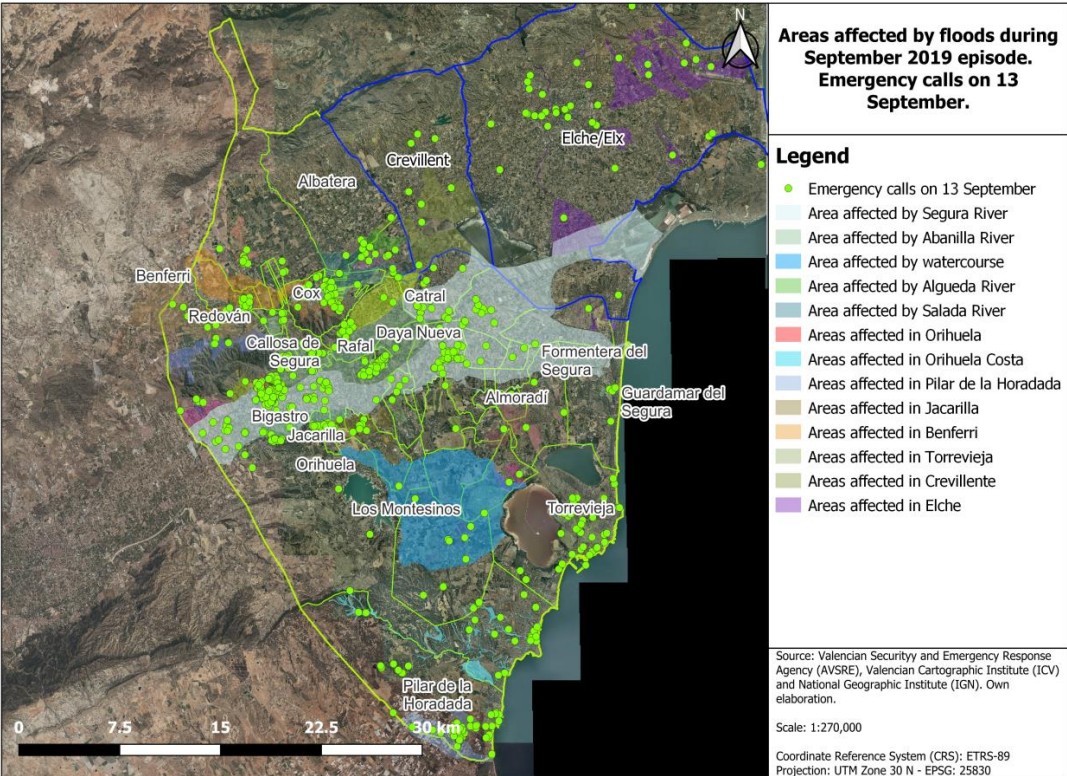

**Figure 10.** Areas affected by floods during September 2019 episode. Emergency calls on 13 September. Source: Own elaboration based on calls to the Emergency Coordination Centre (CCE). AVSRE.

Other towns from where many emergency calls were made are Pilar de la Horadada, Torrevieja, Bigastro and Jacarilla, due to the activation of the channels running close by.

Elche and Crevillente were also affected. The former due to the reactivation of the river Vinalopó, the San Antón ravine, among others, affecting hamlets located in the orchards of Elche. Meanwhile, many calls began to be made from Crevillente due to the entry into operation of the principal ravines.

The next day, it stopped raining. The emergency calls registered were mostly related to floods, the disposal of water or flood rescue. The swollen river Segura caused the rupture of two hillocks in Amoradí, allowing the waters to run across the lowest sector located between the municipalities of Dolores, Daya Nueva, Daya Vieja, San Fulgencio and Guardamar del Segura.

It should be noted that this territory is close to the sea and has hardly any slope. From Almoradí to the Mediterranean Sea, the height above sea level is just five metres, which reduces as the territory approaches the mouth of the river Segura. At this point, it should be pointed out that the flood lasted for a whole month in this spot for several reasons: a lack of a slope and the difficulty for the water to drain, the amount of water that had overflowed, the presence of road infrastructures that acted as barriers, for example, the N-332 and the wall that separates the old channel from the new one of the river Segura.

On 15 September, emergency calls were made in Orihuela and Almoradí. Furthermore, we can observe that calls were also made from the municipalities of Dolores, Daya Nueva and Daya Vieja. It should be highlighted that, contrary to the previous map, there were calls made at the final section of the flooded area, bordering the old course of the river Segura. This justifies and explains the evolution of the overflowed waters. They took two days to reach the mouth of the river and many municipalities were flooded due to the afore-mentioned barriers, hindering the natural drainage of the territory

Figure 11 shows all of the emergency calls made 12–15 September 2019. These calls and their space-time distribution has enabled the elaboration of the map of the flooded areas in the district of Vega Baja del Segura, Elche and Crevillente. As we can observe on the map, the majority of the emergency calls related to floods were made from the large, coloured parts of the flooded areas.

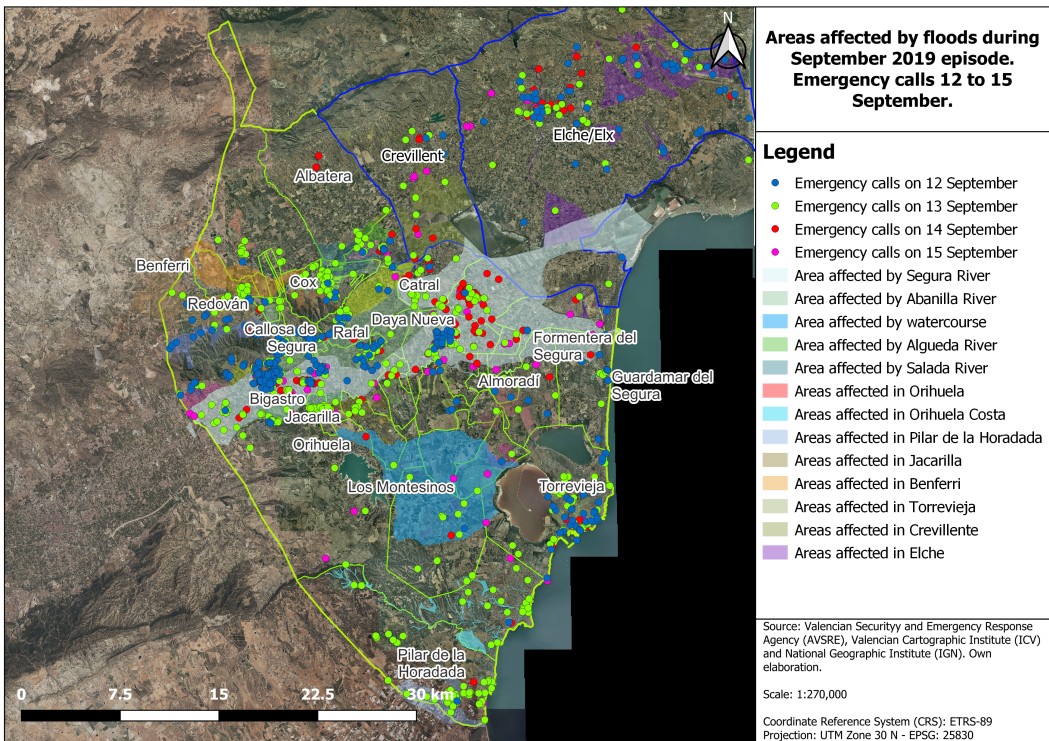

**Figure 11.** Areas affected by floods during September 2019 episode. Emergency calls on 12 to 15 September. Source: Own elaboration based on calls to the Emergency Coordination Centre (CCE). AVSRE.

In order to corroborate this information, a map has been elaborated using the official maps of the Spanish National Cartographic Systems for Flood Areas (SNCZI) associated with a low hazard level, which corresponds to a flood that recurs or with a period of return of 500 years, which is similar to the flood occurring in September 2019.

The map elaborated by the Copernicus satellite has also been used referring to the areas through which water ran in the Vega Baja del Segura. As we can observe in Figure 12, a large part of the Copernicus vectorial map coincides with that of the SNCZI. However, there are areas that the Copernicus identifies as being affected which the official map does not contemplate.

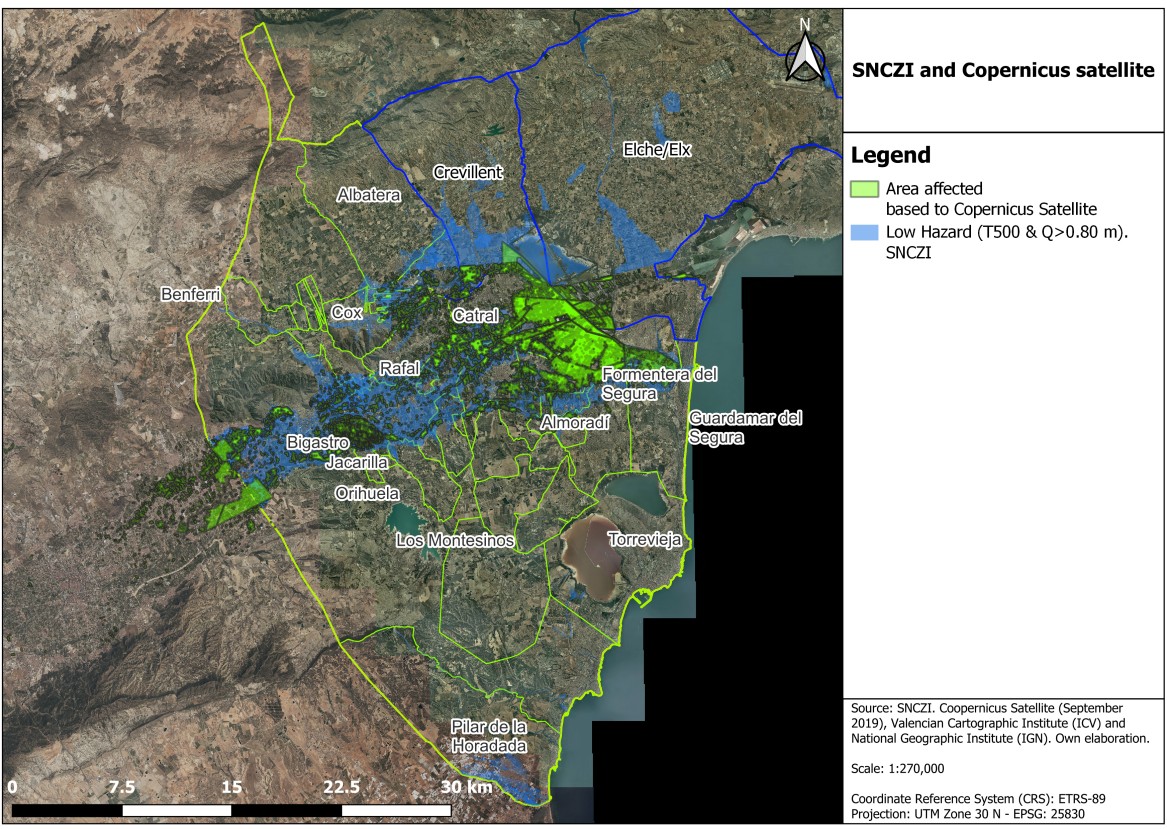

**Figure 12.** SNCZI and Copernicus satellite. Source: SNCZI. Copernicus satellite (September 2019). Valencian Cartographic Institute (ICV) and National Geographic Institute (IGN). Own elaboration.

To all of these maps we can add the emergency calls (112) made on 12 and 13 September which is when the flash floods and most significant effects of the flood episode of September 2019 occurred. As we can see in Figure 12, many of the emergency calls were made from areas marked on both the Copernicus and SNCZI maps. However, there are many other areas which do not appear on the afore-mentioned maps, and the location of these calls allows us to complete them.

It should be noted that many emergency calls were made that do not appear on either the official map or the Copernicus map, particularly in the areas characterised by geomorphological hazard and which generated very high risk situations with major socio-economic damage.

In short, the emergency calls have enabled the mapping of the flood of September 2019, identifying the channels and the overflows that caused it. The detail of this analysis reaches the local level, neighbourhoods and even streets, complementing the official maps so as to determine the real flood risk through the elaboration of a comprehensive risk map.

## 4. Discussion

The results of geolocalising the emergency calls can be represented at any scale. In this case, given the surface area covered by the area of study, we decided to use the district scale which implies the reduction of detailed mapping information. However, it should be indicated that the map obtained can be used at municipal level, by census tracts, neighbourhoods, streets and even homes.

When representing the emergency calls through maps, certain drawbacks can be found. For example, there is a high number of calls for the same incident, for instance, six emergency calls that are represented by just one point due to the information of the database. The transformation of the geographic coordinates (degrees ($°$), minutes ($'$) and seconds ($''$)) to projected coordinates (X, Y) can produce the disappearance of emergency call points or their location in a different point to the one that it corresponds to. Therefore, its position should be modified to its exact point.

The emergency calls have proved to be a fundamental source of data for understanding the functioning of the territory, the identification and type of problems, for identifying spaces that are vulnerable to a natural threat, for undertaking flood mapping and, in short, for designing solutions for the problems detected. Therefore, a field has been opened in terms of applying this information in the geography discipline, which enables us to gain a scalable comprehensive view and learn about the evolution of an event through the space-time distribution of the calls registered in the Emergency Coordination Centre (112). One disadvantage is that this record of calls is only after the catastrophe has happened, although it can be mapped during the emergency.

It would be desirable to be able to process calls during the flood event. This information is only available to the Emergency Coordination Centre (112) when it receives emergency calls, allowing the call to be geolocated. In this way, linked to the reason for the call, the Emergency Coordination Centre is responsible for mobilising the appropriate device to the object of the call. For example, if someone needs to be rescued from their home, the fire brigade or state security forces (police and army) are sent.

However, the present study is carried out after the flood event of 2019, in order to analyse the calls, their reasons, identify the problems, delimit the affected areas and draw up a flood map, which could complement the official mapping, forming an integral flood risk map (Risk: Hazard $\times$ Exposure $\times$ Vulnerability (+ complementary information, for example: 112 or 911 calls)). Knowledge of the real existing problems makes it possible to propose solutions to existing problems that could happen again. This allows for greater preparedness and, consequently, greater resilience in a flood zone, taking advantage of the disaster as an opportunity to do things better.

The analysis of the flood episode of 26–30 November 2016 in the Region of Valencia conducted by the authors Camarasa-Belmonte and Caballero (2018) revealed a series of major problems caused by rain in situ in the district of Vega Baja del Segura. Specifically, a greater impact was observed on the traffic on the A-7, AP-7, N-340, N-332 and streets or avenues in Torrevieja, with just one rainfall recorded between 1 and 50 L/m$^2$ in the different parts of the district, thanks to the information of the emergency calls [22].

The study by Ortiz et al. (2022) also analyses the emergency calls made between 10 and 20 September, which totalled 14,194 calls. Of these, 4078 calls corresponded to the district of Vega Baja del Segura [29]. However, the analysis conducted in this study (12–15 September), analyses a total of 3648 emergency calls made from the area of study. This number of calls is lower as it exclusively takes into account those calls related to weather phenomena and rescue operations, discarding other types of call related to phenomena such as accidents, services, supply-related incidents, etc.

At this point, we should note the importance given to the red alert by AEMET, which led to the cancellation of school and the suspension of some jobs. Without a doubt, the red alert saved lives in the district of Vega Baja and reduced the number of emergency calls that could have been registered both on the roads and in the urban centres. It also avoided unnecessary movements by people who had to collect family members from their

workplaces, schools or residences for dependent people, which led to a reduction in the exposure of the population. If the red alert had not been declared, the social consequences would probably have been greater and the register of emergency calls would have been double or even triple the size.

It has been verified that many of the flooded areas identified through the emergency calls are related to the spaces identified in the environmental report with respect to floods of the Territorial Action Plan (PAT) of Vega Baja del Segura, green infrastructure. New areas have also been incorporated that could be added to the table in the afore-mentioned report [33].

The green infrastructure report of the PAT of Vega Baja includes a comparative table between the flood of November 1987 and September 2019, with its socio-economic effects and the responses implemented after the catastrophe. The Plan Vega Renhace document also includes a comparative map of the two floods focusing on the floodplains of the river Segura [32]. The map obtained through the emergency calls (112) has enabled the elaboration of a much more detailed and precise flood map which complements the information of these two documents, identifying the functioning of the channels which, in general, are not distinguished on the official maps (SNCZI and PATRICOVA). It also enables each agent to determine the zones and identify vulnerable areas.

However, it should be clarified that the comparative flood map does not include all of the floods occurring in the district of Vega Baja. Nor does it contemplate the waters that come from the municipalities of Crevillente and Elche.

With respect to the official maps of Spain (SNCZI) and the Region of Valencia (PATRI-COVA), Olcina et al. (2021) undertook a comparative analysis of the two flood maps, identifying a series of differences with respect to the impact on the surface and buildings [17]. Interestingly, they also indicate that the SNCZI map does not include the floods generated by minor river channels as they are not recognised as Public Water Domain. Meanwhile, the PATRICOVA does include a layer called geomorphological hazard which covers a large part of these types of channels. However, the geomorphological hazard layer has still not been included in the elaboration of the risk map, which only takes into account fluvial hazard. This is a mistake, because a large part of the floods of 2019 occurred in these types of channels.

There were numerous calls were made from towns such as Orihuela, Redován, Callosa de Segura, Cox or Bigastro, which suffer from major floods with the reactivation of these channels which represent a geomorphological hazard. There are even areas, such as San Miguel de Salinas, Los Montesinos, Orihuela costa and Pilar de la Horadada, among others, whose historical channels or paleochannels were reactivated in the flood episode of 2019, in spaces where society did not know that these fluvial channels existed. In some of these cases, the geomorphological hazard layer of the PATRICOVA does not identify these channels. It also enables the official maps to be corrected or complemented. For example, the PATRICOVA indicates that the whole of the municipality of Catral was flooded and assigned it a hazard level of 2. This is wrong as only the orchard area (west, south-west, south, south-east and east) was affected within the municipal limits. This aspect is contemplated in the SNCZI.

All of this information and the impact on the flooded areas by a specific channel enables us to complete and incorporate more detail in the information and the official flood maps. Furthermore, it identifies channels that were reactivated. Therefore, it was able to propose the demarcation of the Public Water Domain, its area of protection and regulation of land uses in the floodable areas. This justifies the need to establish new PWD demarcations in the district of Vega Baja del Segura, Elche and Crevillente.

Thus, the need arises to conduct an in-depth review of the SNCZI and PATRICOVA maps, incorporating the geomorphological hazard layer in the flood risk, showing the real risk of the area of study.

In response to the question posed in the objectives of this article, a large number of calls coincide with the areas contemplated as significant potential risk areas which are

contemplated in the official maps. However, the emergency calls enable us to identify "new" flooded areas which are not contemplated in the official maps but are spaces that are known to be prone to flooding. As a result, the emergency calls better indicate the areas that are really flooded, verifying the spaces affected and the streets and roads that have been cut off, the type of incident reported in the call, the description of the moment and its location.

As a proposal for future research, the use of emergency calls can be applied to the analysis of natural risks and relationships can be established with other elements. For example, one future research line that could be developed is focused on the relationship between emergency calls and the affected roads, damaged homes, affected basements, social or economic losses. They can also be applied to the management of the emergency. They enable a better understanding of how the natural elements in the district function so that a more effective management can be made while saving time for the emergency teams.

Finally, the analysis of the emergency calls reveals the need to conduct a District Flood Emergency Plan, as we can observe in the mapping of the results that floods do not understand administrative boundaries and the reactivation or overflowing of a channel can affect many towns which should know the modus operandi in the case of a flood episode.

## 5. Conclusions

This study shows the applicability of analysing the emergency calls (112) registered in the Emergency Coordination Centres of a territory, allowing to know the evolution of a meteorological phenomenon (heavy rainfall) as well as the evolution of a flooding event in a territory.

The first calls usually correspond to a break in people's normality when heavy rainfall begins to occur. This means that the reasons for the first calls are related to the entry of water into homes or businesses (leaks), disruption of road infrastructures and, indirectly, traffic accidents due to atmospheric adversity.

With the evolution of the flooding episode, depending on the intensity and quantity of rainfall, sporadic river courses begin to be reactivated, while the flow of the main river in the study area (Segura river) begins to increase. With regard to emergency calls, those for rescue or rescue purposes begin to be recorded, as well as calls due to rain and river flooding. These first moments allow us to know the areas most vulnerable to flooding due to the physical conditions of the terrain, in which the most depressed areas or those exposed to a natural hazard begin to be affected.

As the flood event progresses, situations of greater repercussions may arise, such as the overflowing of main watercourses and the flooding of entire population centres, where the population has to be evacuated.

Using the emergency calls enables us to identify fluvial channels which cause serious floods and affect the population, which are not included in the official maps. This enables us to complement them and elaborate a hazard map and, therefore, identify the real flood risk in the area of study.

The use of the emergency calls has shown the possibility of using them for other natural, human or technological risks, and interrelating them with other factors, such as the social and economic impact, the affected roads and the management of the emergency during the event.

In short, the emergency calls have a multidisciplinary application which opens up new fields in scientific research.

**Author Contributions:** Conceptualization, A.O. and J.O.; methodology, A.O.; software, A.O.; formal analysis, A.O. and J.O.; investigation, A.O. and J.O.; resources, A.O.; writing—original draft preparation, A.O. and J.O.; writing—review and editing, A.O. and J.O.; supervision, J.O. All authors have read and agreed to the published version of the manuscript.

**Funding:** This research received no external funding.

**Acknowledgments:** The authors wish to express their gratitude to the Agencia Valenciana de Seguridad y Respuesta a las Emergencias (Valencian Agency for Security and Emergency Response) (AVSRE) and the Centro de Coordinación de Emergencias de la Generalitat Valenciana (Emergency Coordination Centre of the Regional Government of Valencia) (112), for providing the information and data used to conduct this research. In particular, we are grateful to the support provided by Jorge Suárez Torres (Emergencies Sub-director of the Agencia Valenciana de Seguridad y Respuesta a las Emergencias Valencian (Agency for Security and Emergency Response) and José María Ángel Batalla (Director of the Agencia Valenciana de Seguridad y Respuesta a las Emergencias), without which this study could not have been conducted.

**Conflicts of Interest:** The authors declare no conflict of interest.

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
