# Peer review of "Floods and Emergency Management: Elaboration of Integral Flood Maps Based on Emergency Calls (112)—Episode of September 2019 (Vega Baja del Segura, Alicante, Spain)"

_water, doi:10.3390/w15010002_

Round 1

Reviewer 1 Report

The research is interesting but not conclusive. In any case, the number of emergency calls is essential, and the only question is what were the consequences of the rings at the time of the flood event. According to the survey results, calls were only received without special treatment or reaction from the civil protection service. Why data on danger have not been entered into the flood maps already during the flood event? Processing data after the event can be interesting, but it is not as important as if it were processed during the event. The article does not mention the possibilities for such an organization of civil protection.

Author Response

Consulte el archivo adjunto

Reviewer 2 Report

Dear Editor.

I have finished my review on the proposed paper “Floods and emergency management: elaboration of integral flood maps based on emergency calls (112). Episode of September 2019 (Vega Baja del Segura, Alicante, Spain).

Summary of the manuscript:

The article deals with the deputies of the local authorities on floods and how the emergency mapping can be used so as to produce reliable flood maps.

It is an interesting approach and gives some useful insights on the possible usage of the emergency calls for rapid flood mapping. Some issues need to be addressed are the following.

Line 29. Also, state that the aforementioned circumstances lead to an increase number of large-scale studies regarding the flood exposure of critical infrastructure and assets on floods (https://doi.org/10.3390/hydrology9080145, https://doi.org/10.1111/jfr3.12288)

Line 60. Make a reference to the flood directive (2006/60/EC) and how deals with the flood hazard, vulnerability and risk. It is important to mention this guide as it is a common framework for flood risk assessment in European Union .

Figure 1. Exposure is missing from the graph. We cannot have vulnerability without exposure. Vulnerability = Hazard*Exposure.

 Line 154. “Many studies have used emergency calls as additional information in analysing an154 event”… Please cite these article!

Line 154. A similar approach also is the use of twitter post (from official agents). Please state the differences and the advantages of each method.

Figure 2. Please increase the label font size.

Line 353. Better explain the geo-location technique used herein.

Line 905. Exclude this phrase “This section is not mandatory but can be added to the manuscript if the discussion is unusually long or complex”.

Line 914. The Conclusion must not be a repeated abstract. Draw the main conclusion derived from this research

In general the approach is interesting for the journal’s readers and present many novelty parts. To this end, I propose major revision.

Reviewer 3 Report

This is a most interesting paper.  The idea of using emergency call origin locations to trace the pathways of flood water is excellent and the scheme appears to work well.

My comments concern the presentation of the work.  The are difficulties with the English language ansd ways of expressing things. I have made 55 comments on the attached pdf to which I draw your attention.

A major annoying factor is the over reduction of all the maps. I had to use a 300% zoom to be able to read the maps.  Please enlarge then so that all the type faces on all illustrations can be read at A4 page size.

I have given some examples of how sentences can be written more succinctly. Please follow these examples throughout the paper.

Please look again at some of the units you use, particularly lirew per square meter.  That is not conventional in the English-speaking hydrological world.  m3 km2 per unit time is more usual  (or a rainfall depth).

Many statements you have made could be rewritten using fewer words, please try to do so.

Round 2

Reviewer 2 Report

The article in the revised form meets the demands for publication in the WATER journal